

# Crystal structure of the 3C protease from Southern African Territories type 2 foot-and-mouth disease virus

Jingjie Yang[1], Eoin N. Leen[1], Francois F. Maree[2] and Stephen Curry[1]

[1] Department of Life Sciences, Imperial College, London, United Kingdom
[2] Transboundary Animal Disease Programme, Agricultural Research Council, Onderstepoort Veterinary Institute, Onderstepoort, South Africa

## ABSTRACT

The replication of foot-and-mouth disease virus (FMDV) is dependent on the virus-encoded 3C protease (3C$^{pro}$). As in other picornaviruses, 3C$^{pro}$ performs most of the proteolytic processing of the polyprotein expressed from the large open reading frame in the RNA genome of the virus. Previous work revealed that the 3C$^{pro}$ from serotype A—one of the seven serotypes of FMDV—adopts a trypsin-like fold. On the basis of capsid sequence comparisons the FMDV serotypes are grouped into two phylogenetic clusters, with O, A, C, and Asia 1 in one, and the three Southern African Territories serotypes, (SAT-1, SAT-2 and SAT-3) in another, a grouping pattern that is broadly, but not rigidly, reflected in 3C$^{pro}$ amino acid sequences. We report here the cloning, expression and purification of 3C proteases from four SAT serotype viruses (SAT2/GHA/8/91, SAT1/NIG/5/81, SAT1/UGA/1/97, and SAT2/ZIM/7/83) and the crystal structure at 3.2 Å resolution of 3C$^{pro}$ from SAT2/GHA/8/91.

## INTRODUCTION

Diseases caused by RNA viruses are often difficult to control because of the high mutation rate and the continual emergence of novel genetic and antigenic variants that escape from immune surveillance. The degree to which immunity induced by one virus is effective against another is largely dependent on the antigenic differences between them. Foot-and-mouth disease virus (FMDV) is an example of an antigenically variable pathogen that infects many species of cloven-hoofed animals, such as cattle, sheep, pigs and goats, and remains a potent threat to agricultural livestock (*Sutmoller et al., 2003*). Although FMD vaccines made from chemically inactivated virus particles are in widespread use, control of the disease remains difficult. This is because the vaccines provide only short-lived protection and the virus occurs as seven clinically indistinguishable serotypes (O, A, C, Asia1 and three Southern African Territories serotypes: SAT1, SAT2 and SAT3), each of which has multiple, constantly evolving sub-types (*Knowles & Samuel, 2003*). Viruses belonging to the SAT serotypes display appreciably greater genomic and antigenic variation in their capsid proteins compared to serotype A and O viruses (*Bastos et al., 2001*; *Bastos et al., 2003*;

Corresponding author
Stephen Curry,
s.curry@imperial.ac.uk

*Maree et al., 2011*), possibly due to their long term maintenance within African buffalo (*Syncerus caffer*). Constant surveillance of circulating strains is required to ensure that vaccine stocks remain effective.

In common with other members of the picornavirus family, FMDV has a single-stranded, positive-sense RNA genome. Cell entry in infected hosts is followed immediately by translation of a large open reading frame in the viral RNA. This yields a polyprotein precursor of over 2,000 amino acids that is processed into fourteen distinct capsid and non-structural proteins for virus replication. The majority of this processing is done by the virus-encoded 3C protease (3C$^{pro}$), which cleaves the precursor at ten distinct sites. FMDV 3C$^{pro}$ may also assist infection by proteolysis of host cell proteins and has RNA-binding activity that is important for initiation of replication of the viral RNA (reviewed in *Curry et al., 2007b*).

Crystallographic analysis of the 3C$^{pro}$ from a type A FMDV (sub-type A10$_{61}$) showed that, similar to other picornavirus 3C proteases, it adopts a trypsin-like fold consisting of two $\beta$-barrels that pack together to create a centrally-located Cys-His-Asp/Glu catalytic triad in the active site (*Allaire et al., 1994*; *Matthews et al., 1994*; *Mosimann et al., 1997*; *Birtley & Curry, 2005*; *Yin et al., 2005*). Subsequent studies on FMDV 3C$^{pro}$ complexed with peptides derived from the viral polyprotein work revealed that substrate recognition is achieved by conformational changes primarily involving the movement of a $\beta$-ribbon (residues 138-150) that helps to secure the position of cognate peptides in relation to the active site of the protein (*Sweeney et al., 2007*; *Zunszain et al., 2010*).

Sequence analysis has shown that while variation within FMDV 3C$^{pro}$ does not rigidly reflect that observed with capsid proteins, the SAT-type 3C proteases generally form a distinct cluster (*Van Rensburg et al., 2002*). Mapping of the sequence variation between different FMDV serotypes onto the structure of A10$_{61}$ 3C$^{pro}$ indicated that the peptide-binding face of the protease is completely conserved among the non-SAT serotypes (which are 91–97% conserved in amino-acid sequence), supporting the notion that identification of inhibitors of the protease might aid the development of broad spectrum antiviral drugs (*Birtley & Curry, 2005*; *Curry et al., 2007a*). This structure should therefore serve as a useful model for the 3C protease from this group of viruses. However, the same comparison suggested the presence of at least two amino acid differences on the peptide-binding surfaces between A10$_{61}$ 3C$^{pro}$ and the corresponding 3C sequences from SAT serotype viruses.

To provide a more complete picture of the structural variation between FMDV 3C proteases from different serotypes, we set out to determine the crystal structure of 3C$^{pro}$ from at least one SAT serotype virus. We report here the cloning and expression of 3C$^{pro}$ from four distinct SAT1 and SAT2 viruses and the crystal structure of the 3C$^{pro}$ from a SAT2 serotype virus (SAT2/GHA/8/91).

## MATERIALS AND METHODS

### Cloning and mutagenesis

We used the polymerase chain reaction (PCR) to amplify the coding regions for the FMDV 3C proteases of sub-types SAT2/GHA/8/91 (Accession No. AY884136),

**Table 1  DNA primers for cloning and mutagenesis.**

| | SAT2/GHA/8/91 |
|---|---|
| Forward | GATGATCTCGAGGAAGTGGCGCTCCGCCGACCGAC |
| Reverse | CATGCCAAGCTTATGGGTCAATGTGTGCTTTGAGTTGGAGCAGGCTCGACCGTG |
| C142A-for | GGACCAAGGTTGGATACGCTGGAGGAGCCGTCATGAC |
| C142A-rev | GTCATGACGGCTCCTCCAGCGTATCCAACCTTGGTCC |
| C163A-for | CATACAAAGATGTTGTCGTCGCCATGGACGGTGAACACCATGC |
| C163Arev | GCATGGTGTCACCGTCCATGGCGACGACAACATCTTTGTATG |
| | SAT1/NIG/5/81 |
| Forward | GATGATCTCGAGGAAGTGGAGCGCCACCCACCGAC |
| Reverse | CATGCCAAGCTTAAGGGTCGATGTGTGCCTTCATC |
| C142A-for | GCCACCAAAGCTGGTTACGCTGGAGGAGCCGTTCTTG |
| C142A-rev | CAAGAACGGCTCCTCCAGCGTAACCAGCTTTGGTGGC |
| C163A-for | CCTACAAAGACATCGTAGTGGCTATGGATGGTGACACCATGC |
| C163Arev | GCATGGTGTCACCATCCATAGCCACTACGATGTCTTTTGTAGG |
| | SAT1/UGA1/97 |
| Forward | GATGATCTCGAGGAAGCGGTGCGCCACCGACCGAC |
| Reverse | CATGCCAAGCTTATGGGTCGATGTGGGCTTTCATC |
| C142A-for | GGACCAAGGTAGGTTACGCTGGGGCGGCCGTACTGAC |
| C142A-rev | GTCAGTACGGCCGCCCCAGCGTAACCTACCTTGGTCC |
| C163A-for | GTACAACGACATCGTCGTCGCCATGGACGGCGACACCATG |
| C163Arev | CATGGTGTCGCCGTCCATGGCGACGACGATGTCGTTGTAC |
| | SAT2/ZIM/7/83 |
| Forward | GATGATCTCGAGGAAGCGGAGCCCCACCGACCGAC |
| Reverse | CATGCCAAGCTTAAGGGTCGATGTGGGCCTTCATC |
| C142A-for | GGGACCAAAGTTGGATACGCTGGAGCCGCTGTTCTCG |
| C142A-rev | CGAGAACAGCGGCTCCAGCGTATCCAACTTTGGTCCC |
| C163A-for | CCTACAAAGACCTAGTCGTTGCTATGGACGGTGACACCATGC |
| C163Arev | GCATGGTGTCACCGTCCATAGCAACGACTAGGTCTTTGTAGG |

SAT1/NIG/5/81 (Accession No. AY882592), SAT1/UGA/1/97 (Accession No. AF283456), and SAT2/ZIM/7/83 (Accession No. AF540910). In each case the reaction was performed using DNA primers (Table 1) that introduced 5′ *Xho*I and a 3′ *Hind*III restriction sites into the PCR products. These served to facilitate ligation into a version of the pETM-11 vector that had been modified to insert a thrombin cleavage site immediately downstream of the N-terminal His tag (*Birtley & Curry, 2005*). DNA ligations were performed using the Roche Rapid Ligation Kit according to the manufacturer's instructions.

Site-directed mutagenesis was performed with the Quikchange method (Stratagene), using KOD polymerase (Novagen). All DNA sequences were verified by sequencing.

Details of the particular modifications made to expressed proteins are given in the Results and Discussion section.

## Protein expression and purification

All SAT-type 3C proteases were expressed in cultures of BL21 (DE3) pLysS *E. coli* (Invitrogen) grown in lysogeny broth (LB) at 37 °C with shaking at 225 rpm.

Protein expression was induced for 5 h by the addition of 1 mM isopropyl $\beta$-d-1-thiogalactopyranoside (IPTG) once the optical density at 600 nm reached 0.8–1.0. Cells were harvested by centrifugation at 4550 g for 15 min at 4 °C and frozen at −80 °C.

The volumes given below are appropriate for processing the pellet from 1 L of bacterial culture. Cell pellets were thawed on ice and re-suspended in 30 mL Buffer A (50 mM HEPES pH7.1, 400 mM NaCl, 1 mM $\beta$-mercaptoethanol) supplemented with 0.1% Triton X-100 and 1 mM phenylmethylsulfonyl fluoride (PMSF) protease inhibitor. Cells were lysed by sonication on ice and lysates clarified by centrifugation at 29,000 g for 20 min at 4 °C. Protamine Sulphate (Sigma) was added to 1 mg/ml final concentration to precipitate nucleic acids, and lysates were then centrifuged again at 29,000 g for 20 min. The supernatant was filtered using a 1.2 $\mu$m syringe filter and incubated for 90 min at 4 °C with slow rotation in 1 mL bed volume of TALON metal affinity resin (Clontech) pre-equilibrated with buffer A. This slurry was applied to a gravity-flow column and the TALON beads washed three times with 50 mL of Buffer A supplemented with 0, 5 and 10 mM imidazole respectively. His-tagged 3C proteins were eluted in 20 mL of Buffer A containing 100 mM imidazole, followed by a final wash with 10 mL of Buffer A containing 250 mM imidazole. To remove the His tag the eluted protein was mixed with 100 units of bovine thrombin (Sigma) and dialysed for 16 h at 4 °C in 4 L of Buffer A supplemented with 2 mM CaCl$_2$. Cleaved protein was then re-applied to TALON resin to remove the cleaved His tag and other contaminants. The untagged protease was recovered in the flow through, concentrated using Vivaspin concentraters (3 kD MWCO) (Sartorius Stedim Biotech) and further purified by gel filtration using HiLoad 16/60 Superdex 75 gel filtration column (Amersham Bioscience) in Buffer A supplemented with 1 mM EDTA and 0.01% sodium azide at a flow rate of 0.5 mL/min. Peak fractions were pooled, concentrated and stored at −80 °C. Protein concentrations were determined from absorbance measurements at 280 nm using extinction coefficients calculated with the ProtParam tool (*Gastiger et al., 2005*).

## Crystallisation and structure determination

Crystallisation trials with purified SAT-type 3C$^{pro}$ were performed at 4 °C and 18 °C using protein concentrations in the range 5–10 mg/mL. Initial screens were done by sitting drop vapour diffusion using a Mosquito crystallisation robot (TTP Labtech). Typically in each drop 100 nL of protein was mixed with 100 nL taken from the 100 $\mu$L reservoir solution. Trials were performed with the following commercial screens: crystal screen 1 and 2, and PEG/Ion (Hampton Research); Memstart, Memcys, JCSG+, and PACT (Molecular Dimensions); Wizard 1 and 2 (Rigaku Reagents).

Crystals of g3C-SAT2-G(1-208) for data collection were washed in the mother liquor (15% (w/v) PEG-8000, 0.09 M Na-cacodylate pH 7.0, 0.27 M Ca-acetate, 0.01 M Tris pH 8.5, 0.08 M Na-thiocyanate) supplemented with 20% (v/v) glycerol, and immediately frozen in liquid nitrogen in a nylon loop. X-ray diffraction data were processed and scaled with the CCP4 program suite (*Collaborative Computer Project No. 4, 1994*), and phased by molecular replacement using the coordinates of type A10$_{61}$ FMDV 3C$^{pro}$ (PDB ID 2j92; (*Sweeney et al., 2007*)) as a search model in Phaser (*McCoy et al., 2007*). The search model was edited to delete side-chains (to the C$_\beta$ atom) for all residues that differed with

g3C-SAT2-G(1-208) and to remove all the atoms in the $\beta$-ribbon (residues 138-150), since these have been observed to vary in structure between different crystal forms (*Sweeney et al., 2007*). Model building and adjustments were done using Coot (*Emsley et al., 2010*); crystallographic refinement was performed initially with CNS (*Brünger et al., 1998*) and completed using Phenix (*Adams et al., 2010*).

## RESULTS AND DISCUSSION

### Protein expression and crystallisation

We engineered bacterial expression plasmids for FMDV 3C proteases from four SAT sub-types: SAT2/GHA/8/91, SAT1/NIG/5/81, SAT1/UGA/1/97, and SAT2/ZIM/7/83 (see Materials and Methods) which have 80%, 92%, 82% and 85% amino acid sequence identity respectively with the $3C^{pro}$ from FMDV $A10_{61}$ (Fig. 1). In doing so we were guided by the lessons learned from work to express and crystallise subtype $A10_{61}$ FMDV $3C^{pro}$, which suggested that preserving the N terminus of the protein but truncating the C terminus by up to six residues would be optimal for solubility and crystallisation (*Birtley & Curry, 2005*). Accordingly, for each SAT sub-type we generated expression constructs that add a thrombin-cleavable His tag to the N terminus of residues 1-208 of the 213 amino acid 3C protease; following thrombin cleavage there is a single additional Gly residue appended to the N terminus of the protease polypeptide. To ensure the solubility of the SAT-type 3C proteins, we introduced to all constructs a C142A substitution to remove a surface-exposed Cys that had been shown previously to be responsible for protein aggregation (*Birtley & Curry, 2005*; *Birtley et al., 2005*). (The C95K mutation also introduced to eliminate aggregation of $A10_{61}$ FMDV $3C^{pro}$ (*Birtley & Curry, 2005*) was not needed here because residue 95 is an Arg in the SAT 3C proteases used in this study). In addition, the active site nucleophile was eliminated from all constructs by incorporation of a C163A substitution to prevent adventitious proteolysis in highly concentrated samples of purified $3C^{pro}$. For consistency with our earlier naming scheme these SAT2/GHA/8/91, SAT1/NIG/5/81, SAT1/UGA/1/97, and SAT2/ZIM/7/83 3C constructs will be referred to as SAT2/G-g$3C^{pro}$(1-208), SAT1/N-g$3C^{pro}$(1-208), SAT1/U-g$3C^{pro}$(1-208), and SAT2/Z-g$3C^{pro}$ (1-208) respectively.

The $3C^{pro}$ proteins from all four SAT sub-types yielded soluble protein that was purified first by metal-affinity chromatography and then, following thrombin cleavage of the N-terminal His tag, on a gel filtration column (see Materials and Methods). Of the four, SAT1/N-g$3C^{pro}$(1-208) appeared to be the most soluble and could be concentrated to 20 mg/mL (see Table 2). The other three variants exhibited some precipitation during gel filtration, indicated by a void peak containing aggregated $3C^{pro}$, which was about one-third of the area of the monomeric peak. They also had lower apparent solubility limits and could be concentrated to ∼6 mg/mL (SAT2/G-g$3C^{pro}$(1-208)) or ∼11 mg/mL (SAT1/U-g$3C^{pro}$(1-208), and SAT2/Z-g$3C^{pro}$(1-208)).

In crystallisation trials we only obtained crystals from the $3C^{pro}$ of a single sub-type: SAT2/G-g$3C^{pro}$(1-208). These exhibited a variety of habits but the largest were needle-shaped and were typically 10 μm wide and up to 300 μm long. In initial diffraction tests

```
              αN        A1              B1           C1    α1          D1
              hhhhhhhhhssssssss    ssssssssss    sssssshhhh      ssss   s
A10(61)       SGAPPTDLQKMVMGNTKPVELILDGKTVAICCATGVFGTAYLVPRHLFAEKYDKIMLDGR  60
SAT2/GHA/8/91 SGAPPTDLQKMVMANVKPVELILDGKTVALCCATGVFGTAYLVPRHLFAEKYDKVVLDGR  60
SAT1/NIG/5/81 SGAPPTDLQKMVMANTKPVELILDGKTVAICCATGVFGTAYLVPRHLFAEKYDKIMIDGR  60
SAT1/UGA/1/97 SGAPPTDLQKMVMANVKPVELILDGKIVALCCATGVFGTAYLVPRHLFAEKYDKIMLDGR  60
SAT2/ZIM/7/83 SGCPPTDLQKMVMANVKPVELILDGKTVALCCATGVFGTAYLVPRHLFAEKYDKIMLDGR  60
              **.**********.*.************:*****************************:::***

              E1N      E1      E'1   F'1   F1                              A2
              ss    ssssss  sss  ssss  ssssss                         sssssss
A10(61)       AMTDSDYRVFEFEIKVKGQDMLSDAALMVLHRGNCVRDITKHFRDTARMKKGTPVVGVVN 120
SAT2/GHA/8/91 QLDNSDFRVFEFEVKVKGQDMMSDAALMVLNRGQRVRDITMHFRDQVHIKKGTPVLGVIN 120
SAT1/NIG/5/81 AITDRDFRVFEFEIKVKGQDMLSDAALMVLHRGNRVRDITKHFRDQARLRKGTPVVGVIN 120
SAT1/UGA/1/97 ALTNGDFRVFEFEVKVKGQDMLSDAALMVLNRGQRVRDITAHFRDTVRVAKGNPVVGVVN 120
SAT2/ZIM/7/83 ALTDSDFRVFEFEVKVKGQDMLSDAALMVLHSGNRVRDLTGHFRDTMKLSKGSPVVGVVN 120
              : : *:******:*******:********: *: ***:* ****   :: **.**:**:*

                   B2        B'2   C'2   C2              D2      E2
              s   ssssssssssssss ssss  sssss sssss      ssssss   ssssss
A10(61)       NADVGRLIFSGEALTYKDIVVCMDGDTMPGLFAYKAATRAGYCGGAVLAKDGADTFIVGT 180
SAT2/GHA/8/91 NADVGRLIFSGDALTYKDVVVCMDGDTMPGLFAYRAGTKVGYCGGAVMTKDGAHTVIIGT 180
SAT1/NIG/5/81 NADVGRLIFSGEALTYKDIVVCMDGDTMPGLFAYKAATKAGYCGGAVLAKDGAETFIVGT 180
SAT1/UGA/1/97 NADVGRLIFSGDALTYNDIVVCMDGDTMPGLFAYRAGTKVGYCGAAVLTKSGSQTVIIGT 180
SAT2/ZIM/7/83 NADVGRLIFSGDALTYKDLVVCMDGDTMPGLFAYRAGTKVGYCGAAVLAKDGAKTVIVGT 180
              **********:****:*:**************:*.*:.****.**::*.*:.*.*:**

                 E2       F2       αC
              sssss   ssssss   hhhhhhhhhhh
A10(61)       HSAGGNGVGYCSCVSRSMLQKMKAHVEPEPHHE 213
SAT2/GHA/8/91 HSAGGNGVGYCSCVSRSSLLQLKAHIDPEPRTE 213
SAT1/NIG/5/81 HSAGGNGVGYCSCVSRSMLLQMKAHIDPEPHHE 213
SAT1/UGA/1/97 HSAGGNGVGYCSCVSKSMLDQMKAHIDPAPHTE 213
SAT2/ZIM/7/83 HSAGGNGVGYCSCVSRSMLLQMKAHIDPPPHTE 213
              ***************:* *  ::***::* *: *
```

**Figure 1    Amino acid sequence alignment of $A10_{61}$ $3C^{pro}$ with the 3C proteases from the four SAT serotypes used in this study.** Secondary structure features are indicated ($h = \alpha$-helix; $s = \beta$-strand), and coloured and labelled as in Fig. 2B (consistent with the naming scheme used in Birtley et al., 2005).

on beamline ID23-2 at the European Synchrotron Radiation Facility (ESRF) showed that the crystals belonged to a trigonal spacegroup and diffracted to a resolution limit of 2 Å. Unfortunately, for reasons that remain unclear, efforts to reproduce these crystals proved unsuccessful. In subsequent trials diffraction was limited to ∼3 Å.

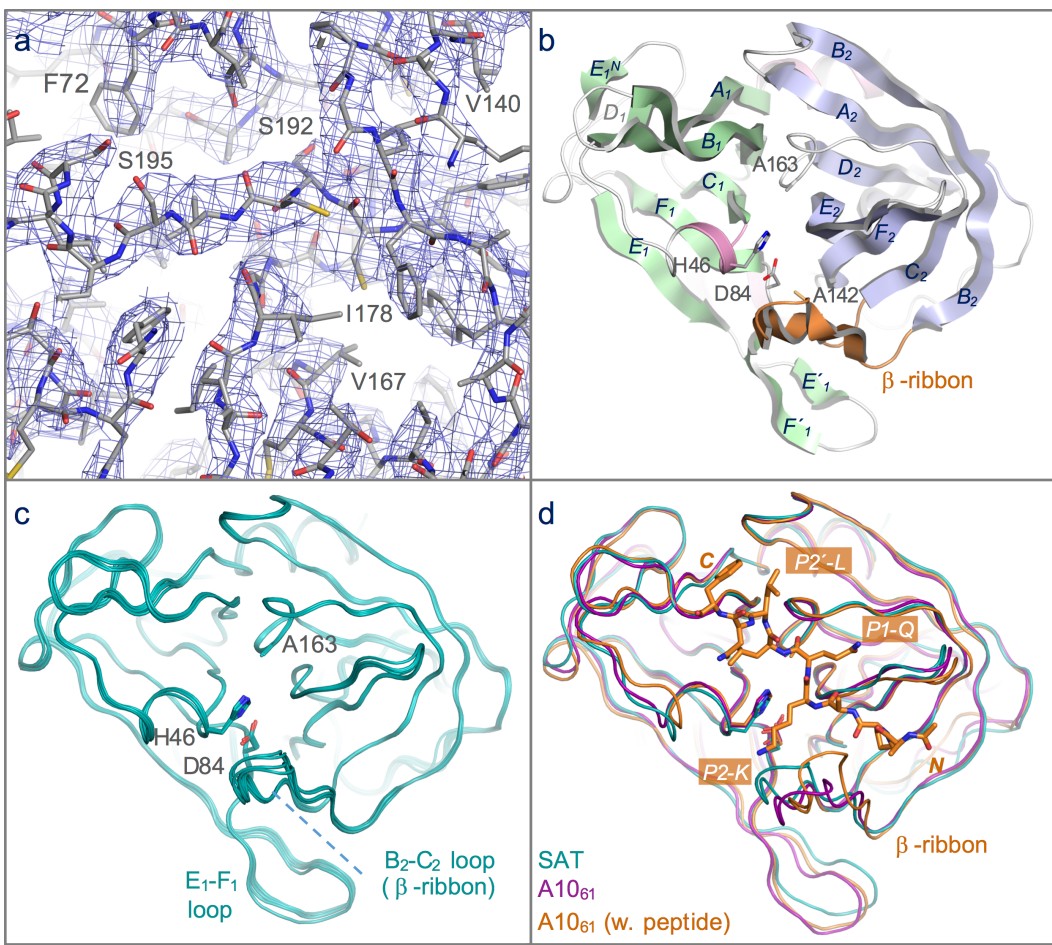

**Figure 2** **Structure of the 3C protease from the SAT2/GHA/8/91 serotype FMDV.** (A) Section of the 3.2 Å resolution electron density map (blue chicken wire) calculated with phases from the final refined model, which is shown as sticks coloured by atom type: grey—carbon; red—oxygen; blue—nitrogen; yellow—sulphur. (B) Overall structure of SAT2/G-g3C$^{pro}$(1-208), with secondary structure elements indicated. The N- and C-terminal $\beta$-barrels are coloured green and blue, respectively. (C) Superposition of the five molecules of SAT2/G-g3C$^{pro}$(1-208) in the asymmetric unit of the crystal, shown in ribbon representation. (D) Comparative superposition of SAT2/G-g3C$^{pro}$(1-208) (teal) with A10$_{61}$ 3C$^{pro}$ in the absence (purple; PDB 2J92) and presence (orange; PDB 2WV4) of a peptide substrate (shown in stick representation).

We used mutagenesis to engineer modifications to the SAT2/G-g3C$^{pro}$(1-208) construct in the search for better crystals. Although alterations to trim the C-terminus by one residue (in SAT2/G-g3C$^{pro}$(1-207)), or to add back a single His residue (in SAT2/G-g3C$^{pro}$(1-207 h))—strategies that had been useful when working with type A10$_{61}$ 3C$^{pro}$ (*Birtley & Curry, 2005*)—both yielded soluble protein (Table 2) and SAT2/G-g3C$^{pro}$(1-207 h) produced crystals, there was no improvement in the resolution of the diffraction.

In a further effort to enhance crystal quality, we used the Surface Entropy Reduction prediction server (*Goldschmidt et al., 2007*) to design additional SAT2/G-g3C$^{pro}$(1-208) mutants. We made four different mutants, each containing the following pairs of

**Table 2 Protein yields and solubilities.**

| Protein | Yield (mg per L of culture) | Maximum concentration (mg/mL) | Aggregation |
|---|---|---|---|
| SAT1/N-g3C$^{pro}$(1-208) | 7.5 | 19.8 | − |
| SAT1/U- g3C$^{pro}$(1-208) | 1.2 | 11.9 | +++ |
| SAT2/Z- g3C$^{pro}$(1-208) | 2.2 | 11.3 | ++ |
| SAT2/G-g3C$^{pro}$(1-208) | 2.5 | 5.7 | ++ |
| SAT2/G-g3C$^{pro}$(1-207h) | 2.1 | 7.2 | + |
| SAT2/G-g3C$^{pro}$(1-207) | 1.7 | 5.6 | +++ |

substitutions: (i) K110T/K111Y (ii) K110Y/K111T; (iii) K51A/K54Y; (iv) K51T/K54S. Of these, only the K51A/K54Y mutant gave protein that was as soluble as wild-type. The K110T/K111Y and K51T/K54S double-mutants produced significantly larger void peaks during purification by gel filtration chromatography, while the K110Y/K111T double-mutant appeared almost entirely aggregated under these conditions. For the three surface-entropy mutants that did yield some soluble protein, no useable crystals were obtained.

### Structure of SAT2/G-g3C$^{pro}$(1-208)

A complete dataset to 3.2 Å resolution was obtained from crystals of SAT2/G-g3C$^{pro}$(1-208). The crystals belong to space-group P3$_2$ and have a long $c$-axis (318.5 Å). The diffraction data were phased by molecular replacement using a search model based on the crystal structure of type A10$_{61}$ FMDV 3C$^{pro}$, which is 80% identical in amino-acid sequence to SAT2/G-g3C$^{pro}$(1-208) (see Materials and Methods). This gave an unambiguous solution with a log likelihood gain of 1495 (*McCoy et al., 2007*), revealing five molecules in the asymmetric unit. Though of modest resolution, the initial electron density maps (Fig. 2A) were of sufficient quality to guide adjustment of the initial molecular replacement model prior to multiple interleaved rounds of refinement and model building. Because of the limited resolution and non-crystallographic symmetry, refinement was performed using group B-factors and non-crystallographic restraints. Model building was done conservatively—amino acid side-chains were truncated to the C$_\beta$ atom in cases where there was no indicative electron density. The final model of SAT2/G-g3C$^{pro}$(1-208) contains residues 7–207 for all five chains and has an $R_{free}$ of 27.2% and good stereochemistry; full data collection and refinement statistics are given in Table 3.

As expected, given the high level of amino acid sequence identity with A10$_{61}$ 3C$^{pro}$ (80%), FMDV SAT2/G-g3C$^{pro}$(1-208) adopts the same trypsin-like fold (Fig. 2B), which has been described in detail elsewhere (*Birtley & Curry, 2005*; *Sweeney et al., 2007*). Superposition of the five molecules in the asymmetric unit shows that they are highly similar to one another (Figs. 1 and 2C)—the pair-wise root mean square deviation in C$_\alpha$ positions between chains is 0.2–0.3 Å. The largest differences are observed in the longest surface-exposed loops, the $E_1$–$F_1$ loop in the N-terminal $\beta$-barrel and the $B_2$–$C_2$ loop known as the $\beta$-ribbon in the C-terminal $\beta$-barrel (Fig. 2C). These are also the regions of greatest difference between SAT2/G-g3C$^{pro}$(1-208) and A10$_{61}$ 3C$^{pro}$; (overlay of the two structures yields an overall

**Table 3   Crystallographic data collection and model refinement statistics for SAT2 3C$^{pro}$.**

| Data collection | |
|---|---|
| Space-group | P3$_2$ |
| a, b, c (Å) | 54.0, 54.0, 318.5 |
| $\alpha$, $\beta$, $\gamma$ (°) | $\alpha = \beta = 90$; $\gamma = 120$ |
| Resolution range (Å) | 53.1–3.2 (3.37–3.2) |
| No. of independent reflections | 17,053 |
| Multiplicity[a] | 2.7 (2.7) |
| Completeness (%) | 99.3 (99.5) |
| $I/\sigma_I$ | 5.7 (1.7) |
| $R_{merge}$(%)[b] | 11.6 (42.4) |
| **Model refinement** | |
| No. of Non-hydrogen atoms | 7,535 |
| $R_{work}$(%)[c] | 22.2 |
| $R_{free}$ (%)[d] | 27.2 |
| Average B-factor (Å$^2$) | 119 |
| RMS deviations—Bonds (Å)[e] | 0.006 |
| RMS deviations—Angles (°) | 1.1 |
| Ramachandran plot (favoured/allowed) % | 89.8/10.2 |
| PDB Accession Code | 5HM2 |

**Notes.**

[a] Values for highest resolution shell given in parentheses.

[b] $R_{merge} = 100 \times \Sigma_{hkl} |I_j(hkl) - \langle I_j(hkl)\rangle\rangle / \Sigma_{hkl} \Sigma_j I(hkl)$, where $I_j(hkl)$ and $\langle I_j(hkl)\rangle$ are the intensity of measurement $j$ and the mean intensity for the reflection with indices $hkl$, respectively.

[c] $R_{work} = 100 \times \Sigma_{hkl} ||F_{obs}| - |F_{calc}|| / \Sigma_{hkl} |F_{obs}|$.

[d] $R_{free}$ is the $R_{model}$ calculated using a randomly selected 5% sample of reflection data that were omitted from the refinement.

[e] RMS, root-mean-square; deviations are from the ideal geometry defined by the Engh and Huber parameters (*Engh & Huber, 1991*).

rms deviation in $C_\alpha$ positions of ~0.6 Å) (Fig. 2D). The flexibility of the $\beta$-ribbon, which shifts in position to aid peptide binding, has been noted before (*Zunszain et al., 2010*) and clearly it plays a similar role in SAT-type 3C proteases.

## CONCLUDING REMARKS

The results reported here provide a template structure of a SAT-type FMDV 3C protease that should be of value in directing molecular investigations of this group of proteases. Although it is frustrating that higher-resolution diffraction data were not obtained, given that initial crystals of SAT2/G-g3C$^{pro}$(1-208) diffracted to 2 Å, this should be possible with further optimization. Likewise, since soluble 3C$^{pro}$ was found to be purified from three other SAT-type viruses—notably SAT1/NIG/5/81—crystal structures for these proteases may well also be achievable.

## ACKNOWLEDGEMENTS

We thank the staff on beamline ID23 at the ESRF for assistance with data collection.

### Funding

This work was supported by the award of a Wellcome Trust studentship to Eoin N. Leen (reference: 083248/2/07/2). The funders had no role in study design, data collection and analysis, decision to publish, or preparation of the manuscript.

### Grant Disclosures

The following grant information was disclosed by the authors:
Wellcome Trust studentship: 083248/2/07/2.

### Competing Interests

The authors declare there are no competing interests.

### Author Contributions

- Jingjie Yang conceived and designed the experiments, performed the experiments, analyzed the data, contributed reagents/materials/analysis tools, wrote the paper, prepared figures and/or tables.
- Eoin N. Leen conceived and designed the experiments, performed the experiments, analyzed the data, contributed reagents/materials/analysis tools, reviewed drafts of the paper.
- Francois F. Maree analyzed the data, contributed reagents/materials/analysis tools, reviewed drafts of the paper.
- Stephen Curry conceived and designed the experiments, performed the experiments, analyzed the data, wrote the paper, prepared figures and/or tables, reviewed drafts of the paper.

### Data Availability

Protein Data Bank
PDB ID 5HM2
http://www.rcsb.org/pdb/explore/explore.do?structureId=5HM2.

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
