# Peer review of "Crystal structure of the 3C protease from Southern African Territories type 2 foot-and-mouth disease virus"

_PeerJ, doi:10.7717/peerj.1964_

## Round 0.1 · original submission · Minor Revisions

· Academic Editor

Minor Revisions

Dear Prof S Curry,

Your review article entitled "Crystal structure of the 3C protease from South African territories type 2 foot-and-mouth disease virus" has now been reviewed, and the reviewer comments are appended below. You will see that they find your work of great interest, but they have raised minor points that should be addressed prior to publication.

We therefore invite you to revise and resubmit your manuscript, taking into account the points raised.

In addition to the general comments of the reviewers, please play attention to these minor comments (added later by reviewer #2)
a) Line 42, it should be “serotypes”
b) Title and Line 43, the correct full name for SAT serotypes is “Southern African Territories” and not “South African Territories”
c) Line 51, there is only one “large” open reading frame but there are others.
d) I think there are 15 different processing products (not 14, line 53) (two forms of L, Lab and Lb).
e) Line 181, I am not sure “troublesome” is the best word here, “unsuccessful” might be better.

Reviewer 1 ·

Basic reporting

In this manuscript, the authors describe the determination of the crystal structure of the 3C protease of a serotype SAT 2 foot-and-mouth disease virus (FMDV). The new structure provides additional information that complements previous information obtained on picornaviral 3C proteases in general, and on the 3C protease of a serotype A FMDV, that had already been determined by the same research group.

Experimental design

The experimental design is fully appropriate with no flaws, as expected from an expert protein crystallography group with a long and successful trajectory in the study of these viral proteases.

Validity of the findings

The authors comment that it was frustrating that higher resolution data could not be obtained despite repeated attempts. Higher resolution data would certainly be helpful in the eventual rational design of antiviral agents able to inhibit the activity of the 3C protease of different FMDV serotypes. The structure obtained at 3.2 angstrom resolution, however, is in itself valid to reveal significant similarities and differences between the 3C protease of two different FMDV serotypes, a result that increases the detailed structural knowledge on this important viral protein.

Reviewer 2 ·

Basic reporting

This manuscript by Jinglie Yang et al., is clearly written and describes the crystal structure of the 3C protease from a SAT2 serotype foot-and-mouth disease virus (FMDV). Previously only the structure of a serotype A FMDV has been determined. As indicated in the Abstract, the SAT serotype viruses are significantly different from the serotype O, A, C and Asia-1 viruses. However, this comparison is usually based on the capsid protein sequences (which define the serotype) and there does not have to be a similar pattern of relationship for other parts of the genome (recombinant genomes can be generated following mixed infection). Indeed phylogenetic analysis by van Rensberg et al (2002) (Gene 289, 19-29, not cited in the current manuscript) suggested quite variable levels of genetic difference between the 3C proteases from SAT viruses.

Experimental design

I think it would be very valuable, for the reader, to provide additional information (other than in the form of the Accession numbers which are included) about the protein sequences for the SAT virus 3C sequences used in this study and that of the serotype A virus studied previously. There is a comment on line 68 indicating 91-97% amino acid conservation but the individual values for each strain are not given and the location of the variation is not indicated. On Line 212, the text indicates that there is a “high level of amino acid sequence identity” between the SAT 2 protein and the serotype A enzyme but this is rather vague and the later mention, on lines 219-220, of particular regions being the regions of greatest difference are not shown anywhere and could usefully be included.

Validity of the findings

It is not clear to me that the concluding sentence (lines 232-234) is valid, it states: “This work has applications in the use of reverse genetic approaches or the design of empty virus capsids to target antigenically significant subtypes in the design of regional vaccines for the control of FMD (Porta et al., 2013)”. Are the authors suggesting that the processing of the SAT FMDV capsid proteins would differ using the 3C proteases from different FMDV serotypes? Is there evidence that the SAT serotype virus 3C proteases differ in their specificity compared to the serotype A 3C?

Additional comments

It should be made clear whether the new structure that has been solved corresponds to a 3C sequence that is 97% identical or 91% identical (or in between) to the serotype A 3C.
I think inclusion of an amino acid sequence comparison also showing the modifications made to improve solubility etc and inactive the enzyme would be helpful.

Reviewer 3 ·

Basic reporting

The article is well writen-in English.
The authors should consider addressing:
1. The intro may benefit elaborating more on FMDV 3Cpro interactions with the viral RNA and host cell proteins published this far
2. Discussion on the structural attributes of SAT 3Cpro in the context of the published knowledge would benefit the manuscript the most, specially, if this work could aid on the design of 3Cpro modifications applicable for FMDV vaccines.

Experimental design

Several modifications were attempted to improve generation of crystals that are of interest to the field. Clarification is needed on the particular modifications introduced in the expression plasmid SAT2/Gg3Cpro (1-208) starting on page 14, Lane 188. Is the N-terminus carrying extra amino acids after these additional modifications?

Validity of the findings

The provided data support the conclusion but this reviewer considers that for clarity the authors should consider:
1) Data on expression, solubility and stability of different SAT 3Cpro are not shown. A table showing the starting material, fold concentration and what specific modification improved protein yield/solubility of mutant/WT 3Cpro are necessary for the purpose of clarification. Likewise an alignment of the FMDV 3Cpro coding sequences that point out at the differences between the two major groups described (but not shown here) would provide support to their statements.
2) add discussions on how the SAT2 proteinase structure potentially relates to the type A proteinase biochemical activity

Additional comments

In the manuscript by Jingjie Yang et al, the authors described the expression, of FMDV 3C proteinases (3Cpro) with the goal of getting crystals suitable for crystallographic analysis. The study focused on the FMDV 3Cpro of SAT serotypes that follows up from the previous studies by the corresponding author on type A 3Cpro published in 2005 (Birthley & Curry and Birtley et al 2005). Here, they expressed four SAT 3Cpros, examined their solubility and tested generation of crystals by these proteins. Next, they described additional modifications introduced on the 3Cpro expression plasmid that resulted on higher solubility and generation of a mutant protein that produced crystals diffracting at a higher resolution (SAT2/Gg3Cpro (1-208). The authors commented on further optimization still needed to produce crystal for these SAT viruses and to draw conclusions on a comparison of SAT 3Cpro to other FMDV serotypes and related viruses.

Overall, this study confirms previous data regarding the properties of FMDV 3Cpro that will be interesting to the virology field.
I have comments and suggestions that the authors should consider addressing as listed above

---

## Round 0.2 · accepted · Accept

· Academic Editor

Accept

I am pleased to tell you that your work has now been accepted for publication in PeerJ.